# Comparison between Colistin and Polymyxin B in the Treatment of Bloodstream Infections Caused by Carbapenem-Resistant *Pseudomonas aeruginosa* and *Acinetobacter baumannii-calcoaceticus Complex*

**DOI:** 10.3390/antibiotics12081317

**Published:** 2023-08-15

**Authors:** Rebeca Carvalho Lacerda Garcia, Rodrigo Douglas Rodrigues, Ester Carvalho Lacerda Garcia, Maria Helena Rigatto

**Affiliations:** 1Medical Sciences Post-Graduation Program, Universidade Federal do Rio Grande do Sul, Porto Alegre 90035-003, Brazil; rebeca.lacerda@hospitaldecaridade.com.br; 2Healthcare-Associated Infection Control Service, Hospital Universitário Professor Polydoro Ernani de São Thiago, Universidade Federal de Santa Catarina, Florianópolis 88036-800, Brazil; rodrigo.douglas@ebserh.gov.br; 3Medical School, Pontifícia Universidade Católica do Paraná, Curitiba 80215-901, Brazil; ester.lacerda@pucpr.edu.br; 4Internal Medicine Department, School of Medicine of the Federal University of Rio Grande do Sul, Porto Alegre 90035-903, Brazil; 5Infectious Diseases Service, Hospital de Clínicas de Porto Alegre, 2350 Ramiro Barcelos St, Porto Alegre 90035-903, Brazil

**Keywords:** bloodstream infection, *Pseudomonas aeruginosa*, *Acinetobacter baumannii*, polymyxin B, colistin

## Abstract

Polymyxins are still widely used for the treatment of carbapenem-resistant *Acinetobacter baumannii* and *Pseudomonas aeruginosa* bloodstream infections (BSIs). This study seeks to evaluate the impact of polymyxin B versus colistin on mortality and nephrotoxicity in BSI caused by these bacteria. We conducted a retrospective cohort study from 2014 to 2021 in Porto Alegre, Brazil. We included patients aged ≥18 years and excluded patients with polymicrobial infection or treatment for ≤48 h. The 30-day mortality was the primary outcome evaluated through Cox regression. We included 259 patients with BSI episodes: 78.8% caused by *A. baumannii* and 21.2% caused by *P. aeruginosa*. Polymyxin B did not impact mortality compared to colistin (adjusted hazard ratio (aHR), 0.82; 95% confidence interval (CI), 0.52–1.30; *p* = 0.40 (when adjusted for COVID-19 comorbidity, *p* = 0.05), Pitt bacteremia score, *p* < 0.01; Charlson comorbidity index, *p* < 0.001; time to start active antimicrobial therapy, *p* = 0.02). Results were maintained in the subgroups of BSI caused by *A. baumannii* (aHR, 0.92; 95% CI, 0.55–1.54; *p* = 0.74), *P. aeruginosa* (aHR, 0.47; 95% CI, 0.17–1.32; *p* = 0.15) and critical care patients (aHR, 0.77; 95% CI, 0.47–1.26; *p* = 0.30). Treatment with polymyxin B or colistin did not impact 30-day mortality in patients with carbapenem-resistant *A. baumannii* or *P. aeruginosa* BSI.

## 1. Introduction

Carbapenem-resistant Gram-negative bacillus bloodstream infections (BSIs) are a worldwide concern, leading to high morbidity, mortality and health-related costs [1,2]. Especially in underdeveloped countries, these infections have a high incidence, becoming a priority in the development of new research [3]. *Acinetobacter baumannii-calcoaceticus complex* (hereafter referred to as *A. baumannii*) and *Pseudomonas aeruginosa* are non-fermentative Gram-negative bacilli frequently associated with serious nosocomial infections, with the pulmonary site being the most affected [4]. Although *A. baumannii* and *P. aeruginosa* are less prevalent than Enterobacterales among Gram-negative BSIs, these pathogens are particularly concerning due to the high rates of multidrug resistance and related mortality [5].

While new drugs such as cefiderocol and ceftolozane–tazobactam show favorable prospective for the treatment of non-fermentative Gram-negatives, their availability is still limited, and their use may be cost-prohibitive in developing countries [6]. Moreover, especially for *A. baumannii*, multiple mechanisms of antimicrobial resistance may play a role in restricting the number of effective therapeutic options. Therefore, the use of polymyxins is still frequent and necessary [4].

Polymyxins (polymyxin B and colistin) are antimicrobials that have re-emerged in recent years as a rescue alternative for the treatment of patients with infections caused by carbapenem-resistant bacteria. Both have a similar mechanism of action, despite differences in their chemical composition and pharmacokinetics [4]. While colistin is administered as sodium colistimethate (CMS)m an inactive prodrug that is converted to colistin in vivo, polymyxin B is administered in its active form, polymyxin B sulfate, assuring faster achievement of the desired plasma levels. CMS is excreted mainly in the renal system, while polymyxin B is excreted mainly by non-renal means [7,8,9]. These pharmacokinetic differences provide some theoretical advantages for polymyxin B over colistin in BSI treatment, while colistin may be preferred in urinary tract infections due to higher local concentrations resulting from the conversion of CMS to colistin in the urine [10].

Despite several studies evaluating the toxicity and pharmacokinetics of polymyxins, few studies have adequately compared the clinical efficacy and impact on mortality between polymyxins. To date, most studies have not been designed to assess mortality as a primary outcome, and many have not selected participants with microbiologically confirmed infections and available minimum inhibitory concentration (MIC) data [11]. Furthermore, there is still a gap in the literature regarding studies exclusively evaluating bloodstream infections caused by *A. baumannii* and *P. aeruginosa*.

This study aims to compare treatment with polymyxin B and colistin, evaluating mortality, microbiological eradication and nephrotoxicity in BSIs caused by carbapenem-resistant *A. baumannii* and *P. aeruginosa*. 

## 2. Results

We evaluated 975 patients with BSI caused by *A. baumannii* or *P. aeruginosa*, among which 259 were included in the final analysis (Figure 1). Of these, 141 (54.2%) were male, with a mean age of 58.5 ± 15.1 years. *A. baumannii* was isolated in 204 (78.8%) of the episodes, and *P. aeruginosa* was isolated in 55 episodes (21.2%). The median Pitt score was 6 (IQR, 1–8), and 184 (71%) patients were admitted to the intensive care unit (ICU) at baseline. Respiratory tract infections were the most common (135, 52.9%), followed by catheter-related infections (48, 18.5%). Of the 48 patients with catheter-related BSI, the catheter was removed within 48 h of the BSI in 38 patients (79.1%).

Two hundred and twelve patients (81.9%) received polymyxin B treatment, compared to 47 (18.1%) who received colistin treatment. The median daily dose was 2,000,000 UI (1,500,000–2,000,000) and 9,000,000 IU (7,000,000–9,000,000) for polymyxin B and colistin, respectively. The loading dose was administered to 96 (37.1%) patients: 68 (32.1%) of 212 from the polymyxin B group compared to 28 (59.6%) of 47 in the colistin group (*p* ≤ 0.001). The median polymyxin MIC value was 0.38 mg/L (IQR, 0.25–0.50). Combination therapy was prescribed for 205 (79.2%) of the infections, whereas a combination treatment with an active antimicrobial was prescript in only 17 (6.6%) of the cases. The most frequent antimicrobial combination therapies were with meropenem (70%) and amikacin (21.2%) (see Appendix A). The general characteristics of the cohort and a univariate analysis comparing baseline variables between the polymyxin B and colistin groups are presented in Table 1.

### 2.1. Primary Outcome

Thirty-day mortality occurred in 129 (49.8%) of the 259 patients: 102 (48.1%) versus 27 (57.4%) patients treated with polymyxin and colistin therapy, respectively (*p*= 0.40). In the multivariable model (Table 2), the antimicrobial therapy (polymyxin B or colistin) did not have a statistically significant impact on 30-day mortality (adjusted hazard ratio (aHR), 0.82; 95% confidence interval (CI), 0.52–1.30; *p* = 0.40). Independent risk factors for 30-day mortality were COVID-19 comorbidity (aHR, 1.54; 95% CI, 1.0–2.35; *p* = 0.05), Pitt score (aHR, 1.09; 95% CI, 1.03–1.16; *p* < 0.01), Charlson comorbidity index (aHR, 1.12, 95% CI 1.05–1.20, *p* < 0.001) and time to start active antimicrobial therapy (aHR, 0.86; 95% CI, 0.76–0.98; *p* = 0.02). The adjusted survival curve is shown in Figure 2. We made an alternative model, holding MIC in the final analysis despite a *p* value > 0.05; however, that did not significantly change the final results with respect to the impact of polymyxin B on mortality (aHR, 0.86; 95% CI, 0.52–1.43; *p* = 0.56). In a sensitivity analysis, polymyxin B did not show a statistically significant difference relative to colistin either in patients whose isolated bacteria MIC was determined by e-test (aHR, 1.6; 95% CI, 0.23–12.4; *p* = 0.60) or in patients for whom broth microdilution was performed (aHR, 0.48; 95% CI, 0.52–1.39; *p* = 0.51). We also performed a sensitivity analysis considering hospital admission, and no statistically significant difference was shown between polymyxins when stratified by this variable (aHR, 1.21 95%; CI, 0.76–1.92; *p* = 0.42)

In the PS-adjusted model, polymyxin B did not show a statistically significant impact on mortality when compared to colistin treatment (aHR, 0.93; 95% CI, 0.57–1.53; *p* = 0.78).

### 2.2. Secondary Outcomes

#### 2.2.1. In-Hospital Mortality

In-hospital mortality occurred in 163 (62.9%) of the included patients: 134 (63.2%) of 212 in the polymyxin B group and 29 (61.7%) of 47 in the colistin group (*p* = 0.87). The antimicrobial therapy (polymyxin B or colistin) did not have a statistically significant impact on in-hospital mortality (aHR, 0.75; 95% CI, 0.49–1.16; *p* = 0.20) when controlled for COVID-19 comorbidity, Charlson comorbidity index; Pitt bacteremia score and time to start active antimicrobial therapy. 

#### 2.2.2. Acute Kidney Injury

Of patients with bacteremia due to *A. baumannii* or *P. aeruginosa*, 79 (30.5%) were not evaluated for acute kidney injury (AKI) because they were already undergoing hemodialysis or had serum creatinine ≥ 4.0 mg/dL on the first day of antimicrobial treatment. AKI occurred in 96 (53.6%) of 179 patients during treatment with polymyxins: 82 (55.0%) of 149 patients and 14 (46.7%) of 30 patients in the polymyxin B and colistin therapy groups, respectively (*p* = 0.43).

Patients who developed AKI were determined according to the Risk, Injury, Failure, Loss, End-Stage (RIFLE) score [12]. Among 149 patients evaluated in the polymyxin B group, 67 (45.0%) did not develop AKI during treatment, 21 (14.1%) patients were classified into the risk group, 19 (12.8%) patients were classified into the injury group and 42 (28.2%) patients were classified into the failure group. Among the 30 patients evaluated in the colistin group, 16 (53.3%) did not develop AKI, 2 (6.7%) were classified into the risk group, 3 (10.0%) were classified into the injury group and 9 (30.0%) were classified into the failure group (*p* = 0.65). Antimicrobial therapy (polymyxin B or colistin) did not have a statistically significant impact on the overall time to AKI occurrence (aHR, 0.98; 95% CI, 0.54–1.80; *p* = 0.95) when controlled for COVID-19 comorbidity, Charlson comorbidity index, Pitt bacteremia score and time to start active antimicrobial therapy. 

#### 2.2.3. Microbiological Clearance

Blood cultures were collected from 182 patients within 30 days after the first bacteremia episode. The indication for the collection of blood cultures was defined by the attending physician. The median time to collect control blood cultures was of 7.5 days (4–13): 8 (4–13) days in the polymyxin B therapy group and 5 (3–10) days in the colistin therapy group (*p* = 0.12). Of these 182 blood cultures, 48 (26.4%) recovered the same bacteria: 40 (26.3%) of 152 in the polymyxin B therapy group versus 8 (26.7%) of 30 in the colistin therapy group (*p* = 0.99).

### 2.3. Subgroup Analyses

Preplanned subgroup analyses were performed for bacteria and baseline ICU admission. The complete results are presented in Appendix A.

#### 2.3.1. *A. baumannii* Infections

In the subgroup of 204 (78.8%) patients with *A. baumannii* infections, 102 (50.0%) patients died in 30 days: 80 (48.5%) of 165 versus 22 (56.4%) of 39 in the polymyxin B and colistin therapy groups, respectively (*p* = 0.48). Antimicrobial therapy (polymyxin B or colistin) did not have a statistically significant impact on 30-day mortality (aHR, 0.92; 95% CI, 0.55–1.54; *p* = 0.74) when adjusted for Pitt bacteremia score, Charlson comorbidity index, COVID-19 comorbidity and time to start active antimicrobial therapy.

#### 2.3.2. *P. aeruginosa* Infections

In the subgroup of 55 (21.2%) patients with *P. aeruginosa* infections, 27 (49.1%) patients died in 30 days: 22 (46.8%) of 47 vs. 5 (62.5%) of 8 in the polymyxin B and colistin therapy groups respectively (*p* = 0.46). Antimicrobial therapy (polymyxin B and colistin) did not have a statistically significant impact on 30-day mortality (aHR, 0.47; 95% CI, 0.17–1.32; *p* = 0.15) when adjusted for Pitt bacteremia score, Charlson comorbidity index, COVID-19 comorbidity and time to start active antimicrobial therapy.

#### 2.3.3. Critical Care Patients

In the subgroup of 184 (71.0%) patients admitted to the ICU at baseline, 108 (58.7%) died in 30 days: 83 (56.8%) of 146 and 25 (65.8%) of 38 in the polymyxin B and colistin groups, respectively (*p* = 0.36). Polymyxin B treatment did not impact 30 day mortality when adjusted for the same variables of the main model (aHR, 0.77; 95% CI, 0.47–1.26; *p* = 0.30).

### 2.4. Post Hoc Power Analysis

As the study did not reach the expected number of inclusions, we performed a post hoc power calculation. Our study had a power of 73.8% to demonstrate the intended difference between groups.

## 3. Discussion

In our study, we evaluated 259 BSIs caused by carbapenem-resistant *P. aeruginosa* or *A. baumannii* treated with polymyxins. We did not find any statistically significant difference in 30-day mortality in patients treated with polymyxin B or colistin when controlled for COVID-19 infection, Charlson comorbidity index, Pitt bacteremia score and time to start active antimicrobial therapy. Although the finding of a longer time to start antibiotics being protective in terms of mortality seems counterintuitive, it probably reflects the fact that large-spectrum antibiotics are commonly administered earlier to patients with more severe infections. In accordance with the main results, the subgroup analysis showed no survival benefit of either polymyxin when analyzing patients with *A. baumannii* and *P. aeruginosa* infections separately or in critical care patients. As secondary outcomes, in-hospital mortality and occurrence of AKI were similar between groups. 

Polymyxins are antimicrobials that show activity in response to most MDR Gram-negatives, especially carbapenem-resistant *P. aeruginosa* (99.6%) and *A. baumannii* (97%), according to SENTRY data [5]. Both polymyxin B and colistin act on the external and cytoplasmic membrane, promoting rapid bacterial eradication, in addition to showing activity against bacterial endotoxins and reducing the expression of TNF-alpha and IL-6 [7,13]. Although their mechanism of action is very similar, expressive pharmacodynamic differences have raised questions as two whether there could be advantages of choosing either polymyxin B or colistin depending on the infection site [8]. In patients with preserved renal function, about 70% of the administered CMS is excreted unchanged in the urine, while only 20–30% is converted to colistin. Renal function has a significant impact on the pharmacokinetics of CMS, and even with high doses, adequate exposure to the antibiotic is not achieved in a significant number of critically ill patients with creatinine clearance greater than 80 mL/min due to the excretion of a large proportion of the CMS before conversion to colistin [9]. Pharmacokinetic studies in humans have shown that colistin takes an average of 36–48 h to reach therapeutic serum concentrations in the absence of administration of an initial loading dose. On the other hand, the pharmacokinetics of polymyxin B are not influenced by renal function [13]. Studies in critically ill patients suggest that the reabsorption rate of polymyxin B in the tubular system is 90–95% and that its clearance is mainly achieved by non-renal systems [10]. Based on pharmacokinetic/pharmacodynamic studies, polymyxin B is currently preferred for critically ill patients due to the unpredictability of achieving reliable serum levels of colistin in its active form during CMS administration [14]. International consensus has even described a preference for the use of polymyxin B in the treatment of critically ill patients [15]. Nevertheless, there is still no strong evidence from clinical studies supporting the choice of one versus the other. 

Previous studies have included comparative analysis of therapeutic efficacy and mortality between polymyxin B and colistin. Oliveira et al. performed a retrospective cohort study evaluating 82 cases of *A. baumannii* infections treated with polymyxins, mostly from BSIs. In that study, the authors found no significant differences in 30-day mortality [16]. Three other cohort studies that were primarily designed to evaluate AKI in patients treated with polymyxin B and colistin also compared mortality risk between these drugs. Most patients had respiratory tract infections, and cases of empiric therapy were also included. No impact on mortality rates was found between polymyxin B and colistin in these studies [17,18,19]. Vardaskas et al. conducted a meta-analysis, compiling data from these previous studies, and did not identify a significant difference in mortality among patients treated with polymyxin B or colistin (RR = 0.71; 95% CI, 0.45–1.13; *p* = 0.99). There was considerable heterogeneity in the etiological agents causing these infections, and none of these studies evaluated polymyxin MIC, which might have introduced bias in the analysis, as we cannot be sure that polymyxins were active in all infections [11]. More recently, a retrospective cohort study of ICU-admitted patients, mostly with respiratory tract infections caused by carbapenem-resistant *Klebsiella pneumoniae*, compared with 68 patients treated with polymyxin B and 36 with colistin, found no significant difference in terms of clinical success or mortality between these groups [20]. 

In our study, we attempted to overcome some of the limitations of the previous data available in the literature to achieve improved reliability when analyzing mortality. We evaluated only BSIs, assuring that patients had clinically relevant infections. Moreover, we included only *A. baumannii* and *P. aeruginosa* infections to achieve a more homogeneous sample, understanding that polymyxins still play a major role in the treatment of non-fermentative bacilli when compared to Enterobacterales. The MIC values of all isolates were tested, the results of which were controlled for. 

Nephrotoxicity is the main adverse effect of polymyxins and should also be considered when deciding whether to prescribe polymyxin B or colistin. A recent meta-analysis of studies conducted using the RIFLE criteria for AKI in polymyxin-treated patients found a pooled incidence of colistin-induced nephrotoxicity of about 48% (95% CI: 42–54), which WAS 10% higher than for polymyxin B (38%; 95% CI: 32–44; RR = 1.37; 95% CI: 1.13–1.6) [21]. Another systematic review and meta-analysis of the topic found an overall rate of polymyxin-induced nephrotoxicity of 39.1%, without statistically significant differences between colistin and polymyxin B. However, pairwise meta-analysis across all studies directly comparing these drugs found higher AKI risk in the colistin-treated patients [22]. In our study, we found an overall AKI rate of 53.6%, which is high compared to the two meta-analysis results reported above, with alarming rates of roughly 30% renal failure in both groups. This result might be explained by the clinical severity of our patients (71.2% were critical care patients, and 43.1% were in septic shock) and concomitant use of other nephrotoxic drugs (22.4% received concomitant aminoglycosides). We did not find differences between colistin and polymyxin regarding AKI risk; however, our study was not primarily designed for this outcome, so we did evaluate important confounding factors, such as the use of other nephrotoxic agents (e.g., contrast media, vancomycin or amphotericin). Nevertheless, our data confirm the development of nephrotoxicity as an important and frequent adverse event of polymyxin therapy. 

This study is subject to some limitations that must be acknowledged. The first and main limitation is the relatively low number of patients included, especially in the colistin group, which decreased the statistical power to detect differences between these drugs. Therefore, we cannot rule out that smaller differences in the mortality rates between these drugs might not have been captured by our model. Nevertheless, if uncaptured differences do exist, they are probably not of considerable magnitude from the perspective of clinical outcomes. Secondly, we highlight that the prevalence of COVID-19 in patients in the colistin group was greater than that in polymyxin B group, which might have been a confounding variable in this analysis. We attempted to minimize its impact by controlling for it in the multivariable model of mortality. Third, the difficulty in estimating dose equivalence between polymyxins (especially in patients with renal dysfunction) and the heterogeneity of antimicrobial combinations limited our comparison regarding these treatment aspects. Nevertheless, median doses used in both groups were adequate according to current recommendations [15]. Fourth, patients included in our study had very severe infections (most were ICU-admitted, with a high proportion in septic shock), which can lead to high mortality rates, regardless of the prescribed antimicrobial therapy. However, we understand that, unfortunately, this is a real-life scenario that reflects the challenge of treating these patients. Finally, due to the retrospective design of the study, it is possible that unmeasured confounding variables may have influenced the analysis. We understand that it is unlikely that large randomized clinical trials comparing polymyxin B with colistin will be available in the near future, considering that most research is focusing on the development of other more attractive antimicrobial options. Therefore, observational studies are still valuable to advance knowledge of this matter.

In summary, in this study, we did not detect differences in 30-day mortality in patients with BSI caused by *A. baumannii* or *P. aeruginosa* treated with polymyxin B or colistin. Larger studies are necessary to confirm these findings. However, at this point, no survival benefit was shown comparing these drugs, despite their relevant pharmacokinetic differences. 

## 4. Materials and Methods

### 4.1. Study Design and Settings

We conducted a retrospective cohort study from 2014 to 2021 in two tertiary-care teaching hospitals in Porto Alegre, Brazil: one with 335 beds and the other with 836 beds. 

### 4.2. Inclusion and Exclusion Criteria 

We included patients aged ≥18 years, with bloodstream infections caused by *A. baumannii* or *P. aeruginosa* resistant to carbapenems treated with polymyxins.

Patients were excluded in the event of death ≤ 48 h after starting antimicrobial treatment, polymyxin treatment duration ≤ 48 h, identification of two or more microorganisms in blood cultures, previous bacteremia episode caused by the same bacteria in <30 days and polymyxin-resistant bacteria.

### 4.3. Variables and Definitions

Our primary outcome was 30-day mortality in patients with *A. baumannii* or *P. aeruginosa*. Secondary outcomes were in-hospital mortality, analysis of blood culture results collected within 30 days of bacteremia and AKI defined according to RIFLE score [12]. Our main independent variable was antimicrobial therapy with polymyxin B or colistin.

Bacterial identification and antimicrobial susceptibility tests were performed using a Vitek2^®^ (bioMérieux, Marcy l’Etoile, France) automatized system in hospital 1 and a matrix-assisted laser desorption ionization–time of flight (MALDI-TOF) system (bioMérieux, Marcy l’Etoile, France) in hospital 2. Results were interpreted according to the Clinical and Laboratory Standards Institute (CLSI) until 2020 [23]. Since 2020, susceptibility results have been interpreted according to the Brazilian Committee on Antimicrobial Susceptibility Testing (BrCAST) [24]. Polymyxin MIC values were determined using gradient tape until April 2018, after which point they were determined using the broth microdilution technique.

We analyzed the medical records of all patients who had positive blood cultures for *P. aeruginosa* or *A. baumannii* between 2014 and 2021, using previously defined inclusion and exclusion criteria. Patients were included in the first day of the BSI, which was defined as the day when the blood sample was drawn. 

Variables potentially related to outcomes were assessed at baseline: demographic variables (age and sex), comorbidities (underlying diseases of patients and the Charlson comorbidity index) [25], COVID-19 diagnosis during hospitalization (before or during the bacteremia episode), primary infection site according to medical staff evaluation (respiratory, abdominal, urinary, skin and soft tissue, catheter-related bloodstream infection or undefined site), ICU admission, invasive mechanical ventilation, hemodialysis, Pitt bacteremia score [26], septic shock, isolated bacteria (*A. baumannii* or *P. aeruginosa*), time to start polymyxins, daily antibiotic dose, antimicrobial combination therapy (defined as the association of two or more antimicrobials, regardless of in vitro activity, started within 48 h of the beginning of polymyxins and that lasted for >48 h), active antimicrobial combination therapy (active antimicrobials were defined as those toward which bacteria had in vitro susceptibility), removal or retention of central venous catheter within 48 h of BSI (in catheter-related infections), baseline renal function and occurrence of AKI during antimicrobial treatment. 

The patient followup period was from the date of diagnosis of BSI to hospital discharge. A preplanned subgroup analysis was performed separately for *A. baumannii* and *P. aeruginosa* and for patients admitted to the ICU during their inclusion in the study.

### 4.4. Sample Size

A sample size of 268 subjects (214 in the polymyxin B group and 54 in the colistin group) was calculated to test whether the hazard ratio for death within 30 days between groups was 0.6, the value chosen considering clinical significance. The probability of death for the polymyxin B and colistin groups by the end of followup was estimated at 50% and 65%, respectively. The calculation considered a power of 80% and a significance level of 5% [27].

### 4.5. Statistical Analysis

Statistical analyses were performed using SPSS for Windows, Version 18.0. Variable distributions were tested by Shapiro–Wilk test. We calculated the median and (p)25th and 75th (p25–p75) percentiles for ordinal or non-normally distributed variables, mean and standard deviation (SD) for normally distributed variables and total and percentage value for categorical variables. Bivariate analysis was performed separately for each of the baseline variables to evaluate differences between polymyxin and colistin therapy groups and factors potentially related to 30-day mortality. *p* values were calculated using Fisher’s exact test for categorical variables and Student’s *t*-test or Mann–Whitney U test for continuous variables. All tests were two-tailed, and a *p* value < 0.05 was considered significant.

A Cox regression model was used to assess the effect of antibiotic therapy on 30-day mortality, adjusting for other potential confounding variables. We chose to use this model so that we could censor patients from the analysis upon hospital discharge or after 30 days (whichever came first), as we did not contact patients afterwards. Variables with *p* < 0.20 in the bivariate analysis were included one by one, in a stepwise-forward model, starting with those with the lowest *p* values. If there were equal *p* values, the variable with the greatest magnitude of effect was included first. Variables with *p* < 0.05 were retained in the model. We also developed a propensity score (PS) using logistic regression for the prescription of polymyxin B or colistin including demographic variables, comorbidities, the hospital to which the patient was admitted, hospitalization time before infection, infection site, ICU admission, need for mechanical ventilation, vasopressor use and Pitt bacteremia score. We further adjusted the impact of polymyxin B (compared to colistin) on 30-day mortality for this score in a Cox regression model.

Secondary outcomes were in-hospital mortality, analysis of blood culture results collected within 30 days of BSI and AKI; the latter was also tested in a Cox regression model, adjusting for the baseline variables that differed between the polymyxin B and colistin groups, following the same criteria as the main multivariable model. We conducted a subgroup analysis separately for *A. baumannii* and *P. aeruginosa* infections and for patients admitted to the ICU during their inclusion in the study. We also performed a sensitivity analysis separately evaluating the impact of MIC values determined using gradient tape and those obtained using the broth microdilution technique on the outcome.

## 5. Conclusions

No statistically significant difference in 30-day mortality was found between treatment with polymyxin B or colistin in patients with BSI caused by *A. baumannii* or *P. aeruginosa*, despite the relevant pharmacokinetic differences of these drugs. Larger randomized studies are necessary to confirm these findings.

## Figures and Tables

**Figure 1 antibiotics-12-01317-f001:**
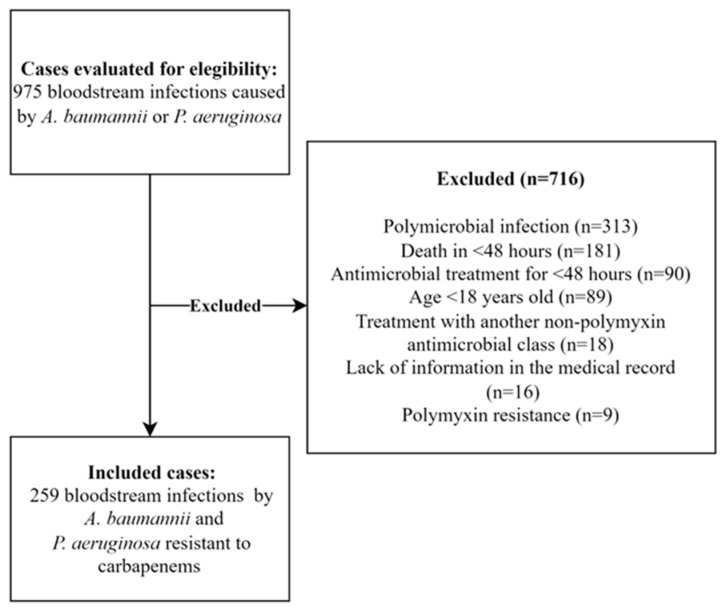
Study inclusion flow chart.

**Figure 2 antibiotics-12-01317-f002:**
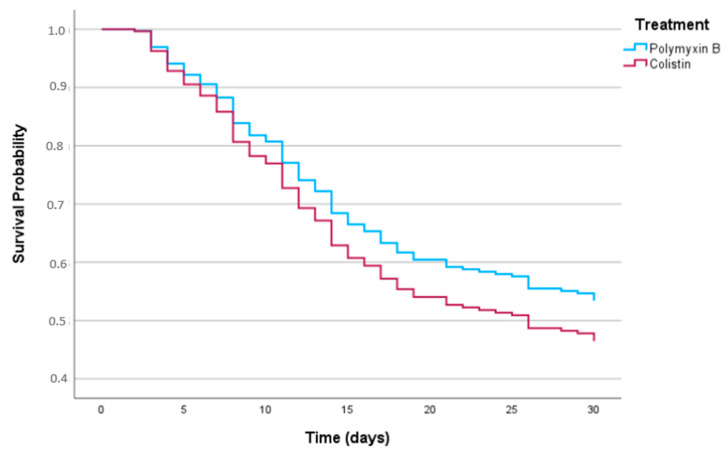
Thirty-day survival curves for patients treated with polymyxin B or colistin adjusted for Charlson comorbidity index, Pitt bacteremia score, COVID-19 infection and time to start active therapy.

**Table 1 antibiotics-12-01317-t001:** Cohort characteristics and univariate analysis of baseline variables according to antimicrobial therapy in patients with *Pseudomonas aeruginosa* or *Acinetobacter baumannii-calcoaceticus Complex* bloodstream infections.

Variable	Total Cohort	Antimicrobial Therapy		
	N = 259	Polymyxin B, N = 212	Colistin, N = 47	*p*
Demographics				
Age	58.5 ± 15.1	59.2 ± 14.9	55.3 ± 15.6	0.11
Weight (kg)	74.3 ± 20.5	73.5 ± 20.3	78.1 ± 21.5	0.16
Sex (masculine)	141 (54.2)	112 (52.8)	29 (61.7)	0.33
Hospital 1	190 (73.4)	164 (77.4)	26 (55.3)	0.003
Hospital 2	69 (26.5)	48 (69.6)	21 (30.4)	0.003
Comorbidities				
Cardiovascular	164 (63.3)	136 (64.2)	28 (59.6)	0.62
Pulmonary	91 (35.1)	73 (34.4)	18 (38.3)	0.62
Neurological	70 (27.0)	64 (30.2)	6 (12.8)	0.02
Hepatic	20 (7.7)	17 (8.0)	3 (6.4)	0.99
Chronic kidney disease	103 (39.8)	85 (40.1)	18 (38.3)	0.87
Gastrointestinal	46 (17.8)	38 (17.9)	8 (17.0)	0.99
Diabetes	81 (31.3)	71 (33.5)	10 (21.3)	0.12
HIV	21 (8.1)	19 (9.0)	2 (4.3)	0.39
Rheumatic	10 (3.9)	7 (3.3)	3 (6.4)	0.40
Oncologic	37 (14.3)	32 (15.1)	5 (10.6)	0.50
Hematological	17 (6.6)	14 (6.6)	3 (6.4)	0.99
COVID-19	79 (30.5)	51 (24.1)	28 (59.6)	<0.001
Infection severity				
Charlson comorbidity index	5 (3–7)	5 (3–7)	4 (2.25–6)	0.09
ICU	184 (71.0)	146 (68.9)	38 (80.9)	0.11
Mechanical ventilation	155 (59.8)	123 (58.0)	32 (68.1)	0.25
Pitt bacteremia score	6 (1–8)	6 (1–8)	6.5 (3.25–8)	0.17
Septic shock	111 (42.9)	87 (41.0)	24 (51.1)	0.25
Microbiological data				
*Acinetobacter baumannii*	204 (78.8)	165 (77.8)	39 (83.0)	0.55
*Pseudomonas aeruginosa*	55 (21.2)	47 (22.2)	8 (17.0)	0.56
Multidrug-resistant	255 (98.5)	209 (98.6)	46 (97.9)	0.55
Polymyxin MIC	0.38 (0.25–0.50)	0.38 (0.25–0.50)	0.50 (0.25–1.0)	0.04
Time from hospitalization to bacteremia (days)	18 (10–33)	19 (10–34)	16.5 (10–22.5)	0.09
Infection site				
Pulmonary	137 (52.9)	110 (51.9)	27 (57.4)	0.52
Urinary	23 (8.9)	23 (10.8)	0 (0)	0.01
Abdominal	20 (7.7)	19 (9.0)	1 (2.1)	0.14
Central venous catheter	48 (18.5)	35 (16.5)	13 (27.7)	0.10
Skin and soft tissues	7 (2.7)	7 (3.3)	0 (0)	0.36
Febrile neutropenia	10 (3.9)	9 (4.2)	1 (2.1)	0.70
Undefined site	27 (10.4)	20 (9.4)	7 (14.9)	0.29
Antimicrobial treatment				
Antimicrobial combination therapy	205 (79.2)	162 (76.4)	43 (91.5)	0.03
Active antimicrobial combination therapy	17 (6.6)	13 (6.1)	4 (8.5)	0.10
Time to start of active antimicrobial therapy (days)	1 (0–2)	1 (0–2)	0 (0–1.75)	<0.01
Loading dose	96 (37.1)	68 (32.1)	28 (59.6)	<0.01

HIV, human immunodeficiency virus; ICU, intensive care unit admission; MIC, minimum inhibitory concentration. Results are presented as: mean ± standard deviation, median (interquartile range) or n (%).

**Table 2 antibiotics-12-01317-t002:** Multivariable analysis for 30-day and in-hospital mortality in patients with *Pseudomonas aeruginosa* or *Acinetobacter baumannii-calcoaceticus Complex* bloodstream infections.

Variable	30-Day Mortality		
	aHR	95% CI	*p*
Antimicrobial therapy (polymyxin B)	0.82	0.52–1.30	0.40
COVID-19 infection	1.54	1.01–2.35	0.05
Time to start susceptible antimicrobial therapy	0.86	0.76–0.98	0.02
Charlson comorbidity index	1.12	1.05–1.20	<0.001
Pitt bacteremia score	1.09	1.03–1.16	<0.01

aHR, adjusted hazard ratio; CI, confidence interval.

## Data Availability

Unidentified datasets will be made available by the corresponding author upon reasonable request.

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
