# Peer review of "Comparison between Colistin and Polymyxin B in the Treatment of Bloodstream Infections Caused by Carbapenem-Resistant Pseudomonas aeruginosa and Acinetobacter baumannii-calcoaceticus Complex"

_antibiotics, 2023, doi:10.3390/antibiotics12081317_

Round 1
Reviewer 1 Report
The manuscript described a comparative study between colistin and polymyxin B treatment of bloodstream infections caused by carbapenem resistant Pseudomonas aeruginosa and Acinetobacter baumannii-calco- aceticus Complex. The study is good but Interest to the readers will be less. Further all organism names should be in Italic (Line No. 74,76, 77). Please check throughout the manuscript. The statistical date should be evaluated by a medical statistician. The discussion part should be improved by adding some latest data.
Minor editing of English language required
Author Response
Reviewer 1
The manuscript described a comparative study between colistin and polymyxin B treatment of bloodstream infections caused by carbapenem resistant Pseudomonas aeruginosa and Acinetobacter baumannii-calco- aceticus Complex. The study is good but Interest to the readers will be less. Further all organism names should be in Italic (Line No. 74,76, 77). Please check throughout the manuscript. The statistical date should be evaluated by a medical statistician. The discussion part should be improved by adding some latest data.
AR: We thank the reviewer for the important comments and suggestions. We double checked the manuscript and corrected all the organism names to italics. Also, sccording to the reviewer’s recommendation, we submitted the article to a medical statistician revision, who suggested the following adjustments:
- A sensitivity analysis for hospital in which the patient was admitted (lines 114-116)
- Including a Survival Curve for the main outcome (added figure 2)
- Changing the term “multivariate” to “multivariable” throughout the text.
- Adding a supplementary table for subgroup analysis with a P value for interaction in each subgroup. (Supplemental Table 2)
- Describing that the propensity score was done by logistic regression (line 362)
- Describing that normality was tested by Shapiro-Wilk test. (lines 346-347)
Finally, we added to the discussion section the reference 20 “Comparative study of polymyxin B and colistin sulfate in the treatment of severe comorbid patients infected with CR-GNB” BMC Infect Dis. 2023 (lines 236 and 239)

Reviewer 2 Report
The article, titled: Comparison between colistin and polymyxin B in the treatment of bloodstream infections caused by carbapenem resistant Pseudomonas aeruginosa and Acinetobacter baumannii-calcoaceticus Complex deals with the comparison of the effect of colistin and polymixin B on carbapenem resistant P. aeruginosa and Acinetobecter baumannii-calcoaceticus blood-stream infections (septicaemia) using the 30 days mortality as an indicator. The authors appropriately designed the study and used several statistics supporting the comparability of the groups of patients and finally they did not find any difference between the effectivity of the two drugs on the 30 days mortality.
Unfortunately this result is not surprising. The limitations of the study, indicated by the authors, serve some explanation, but the patients, most of them in a very serious condition as they were practically in the end-stage period of their diseases, and did not respond to the antibiotics. The authors might find some information whether the clinical condition of the patients showed some improvement after introducing the antibiotics or not. Patients in septic shock with serious underlying conditions (diseases) are generally in an “immunocompromised” situation and require lot of other medication and support. The 30-day mortality occurred as129 (49.8%) of the 259 patients, and 71.2% were critical care patients and 43.1% were in septic shock with many of them acute kidney injury (53.6%).
Some explanation is necessary to the following sentence: “Combination therapy was prescribed for 205 (79.2%) of the infections, however, the combination with an active antimicrobial occurred in only 17 (6.6%) of the cases”. What is the difference between “combination therapy” (combination with other antibiotics or only with other drugs? /cardiac drugs, diuretics?/ and the “combination with an active antimicrobial drugs”?
Author Response
Reviewer 2
1)The article, titled: Comparison between colistin and polymyxin B in the treatment of bloodstream infections caused by carbapenem resistant Pseudomonas aeruginosa and Acinetobacter baumannii-calcoaceticus Complex deals with the comparison of the effect of colistin and polymixin B on carbapenem resistant P. aeruginosa and Acinetobecter baumannii-calcoaceticus blood-stream infections (septicaemia) using the 30 days mortality as an indicator. The authors appropriately designed the study and used several statistics supporting the comparability of the groups of patients and finally they did not find any difference between the effectivity of the two drugs on the 30 days mortality.
Unfortunately this result is not surprising. The limitations of the study, indicated by the authors, serve some explanation, but the patients, most of them in a very serious condition as they were practically in the end-stage period of their diseases, and did not respond to the antibiotics. The authors might find some information whether the clinical condition of the patients showed some improvement after introducing the antibiotics or not. Patients in septic shock with serious underlying conditions (diseases) are generally in an “immunocompromised” situation and require lot of other medication and support. The 30-day mortality occurred as129 (49.8%) of the 259 patients, and 71.2% were critical care patients and 43.1% were in septic shock with many of them acute kidney injury (53.6%).
AR: We thank the reviewer for this comment, and we agree with it. In fact patients had very severe disease, which is a common scenario found in the context of carbapenem-resistant P.aeruginosa or A.baumannii infections. We understand that many times it is difficult to rescue these patients regardless of the antimicrobial therapy prescribed. We added this to the discussion (lines 279-283). Nevertheless, this is a real-life scenario and efforts should continue to be made trying to establish what is the best antimicrobial strategy in these cases.
2) Some explanation is necessary to the following sentence: “Combination therapy was prescribed for 205 (79.2%) of the infections, however, the combination with an active antimicrobial occurred in only 17 (6.6%) of the cases”. What is the difference between “combination therapy” (combination with other antibiotics or only with other drugs? /cardiac drugs, diuretics?/ and the “combination with an active antimicrobial drugs”?
AR: We thank the reviewer for this remark. “Combination therapy” referred to the combination with an antimicrobial regardless of in vitro activity of the drug, for example, including combinations with meropenem. To make it clearer we better described this definition in methods (lines 329-333) and changed “Combination therapy” to “Antimicrobial combination therapy” in table1.
Round 2
Reviewer 2 Report
The corrected form of the article is acceptable. No further corredtion is requested.